# Few Open Access Journals Are Compliant with Plan S

**Jan Erik Frantsvåg** [1,*] **and Tormod Eismann Strømme** [2]

[1]   The University Library of Tromsø, UiT The Arctic University of Norway, NO-9037 TROMSØ, Norway
[2]   University of Bergen Library, University of Bergen, Bergen NO-5020, Norway; tormod.stromme@uib.no
[*]   Correspondence: jan.e.frantsvag@uit.no

**Abstract:** Much of the debate on Plan S seems to concentrate on how to make toll-access journals open access, taking for granted that existing open access journals are Plan S-compliant. We suspected this was not so and set out to explore this using Directory of Open Access Journals (DOAJ) metadata. We conclude that a large majority of open access journals are not Plan S-compliant, and that it is small publishers in the humanities and social sciences (HSS) not charging article processing charges (APC) that will face the largest challenge with becoming compliant. Plan S needs to give special considerations to smaller publishers and/or non-APC based journals.

**Keywords:** Plan S; open access journals; article processing charges (APC); technical requirements; publisher size; Directory of Open Access Journals (DOAJ)

## 1. Introduction

Plan S is an initiative for open access publishing that was launched on 4 September 2018. The plan is supported by cOAlition S, which consists of an international consortium of research funders. The coalition, which by 11 December 2018 consists of 13 national funders and two charitable foundations, is supported by the European Research Council (ERC) and the European Commission [1]. The plan is structured around 10 principles and the main target is to ensure that all research publications funded by the participating funders are published in compliant open access journals or on compliant open access platforms from 1 January 2020 [2]. The members of cOAlition S are also committed to revise the incentive and reward system of science and support the intentions of the San Francisco Declaration on Research Assessment (DORA) [3], which states that journal-based metrics should not be used as a surrogate measure of the quality of individual research articles or individual researchers.

On 27 November 2018, the guidance on the implementation of Plan S [4] was released, clarifying the details for implementation of the initial principles. For scholarly articles to be compliant with Plan S, they must be made openly available immediately upon publication. They must also be published with an open license, limited to Creative Commons Attribution (CC BY), Creative Commons Attribution-Sharealike (CC BY-SA) or dedicated to the public domain (CC0). The guidance lists three ways in which researchers can publish work that is compliant with the plan. First, authors can publish in compliant open access journals or on compliant open access platforms. Second, a peer reviewed version of an article can be deposited in a compliant repository immediately upon publication. Third, authors can publish open access in subscription journals if the journals are covered by a transformative agreement that includes a commitment to transition to open access. The implementation guidance also lists technical requirements and recommendations, for compliant open access journals, platforms and repositories which will be discussed in the following sections.

Plan S has been much debated since its release and has been met with opposition from a number of researchers and publishers. The Norwegian newspaper, *Khrono* (https://khrono.no/), which covers higher education and research, published more than 50 articles and opinions on the topic between

September and December 2018. An open letter expressing concerns over Plan S was published on 5 November 2018 [5]. The letter, signed by more than 1500 people by December 2018, claims that Plan S will limit researchers' freedom to choose publication venues and thus be a serious violation of academic freedom. Another open letter in support of funders' open publishing mandates was later released and has by December 2018 been signed by more than 1,800 people [6]. Traditional publishers have been critical of the plan, claiming that it may undermine the whole research publishing system and that it does not support high quality publishing [7]. Open access publishers have supported the plan and its push for immediate access [8,9].

While most of the debate has been on academic freedom, quality of research publications and the effect on toll access publishers, there have also been discussions on how the plan might affect open access journals and publishers. In a statement of support the Open Access Publishers Association (OASPA) raises the question of how smaller open access publishers, scholarly societies and innovative new publishing platforms may be placed at a disadvantage unless specific provisions are made to include them in centralized funding arrangements [10]. OASPA also questions how resources will be made available to open access publications that do not rely on APC and stresses the importance of supporting a range of business models. Leslie Chan, a long-time open access advocate, says that if the APC model becomes the norm, it will further existing inequality, and points to the need to support a diversity of innovative models and experimentations [11]. Concerns have also been raised about the technical requirements in the implementation guidance on Plan S and how these requirements might affect especially smaller, independent and society-published open access journals [12].

The vast majority of open access publishers are very small and publish only a single journal, while only a few publishers publish 10 or more journals [13,14]. Journals of smaller publishers generally publish fewer articles than journals of larger publishers [15]. As shown in the work by Morrison [14] and Crawford [15], there is a correlation between publisher size, business model and publisher type. Open access publishing is often confused with a "pay-to-publish" model, but 70% of journals registered in DOAJ do not charge author-side fees (APC) [15]. However, these non-APC journals publish less than 50% of all articles. In 2017, 13 publishers accounted for 19.5 % of journals in DOAJ and published 43% of all articles. This group of publishers publishes mostly fee-based journals. Of "no fee" open access journals, Morrison found 99% to have a not-for profit society, university, or government publisher involved, with almost 50% of these journals having a university publishing model [14].

The distribution of journal and publisher size and type and level of APC differs greatly between different subject segments as shown by Crawford [15]. In Biomed, 53% of the journals publish 70% of all articles and charged APCs. The average APC was $1778. In science, technology, engineering, and mathematics (STEM) 40% of journals—publishing 67% of all articles—charged APCs. The average APC was $1507. In the humanities and social sciences (HSS) the journals are mostly small and very few charge APCs. 88% of the journals, publishing 81% of total articles, did not charge APCs. Only 5% of HSS articles were published with fees of $600 or more. While in HSS the smaller journals, i.e., journals publishing below 60 articles yearly, published over 60 percent of all articles, in Biomed and STEM the smaller journals published below 20 percent of all articles. Another variance is type of publisher dominating the different segments. While most of the articles in HSS are published by universities, colleges or institutes, (professional) open access publishers dominate article output in Biomed and STEM.

Publishing entails costs and there are many different business models in open access publishing. (see http://oad.simmons.edu/oadwiki/OA_journal_business_models for an overview). The largest publishers in DOAJ mainly charge APCs, while a sponsorship or subsidy model covers many of the university published journals. From the start of open access publishing, many journals have been heavily reliant on voluntary work [16]. This feature is still relevant with many small open access journals operating on very low budgets, with editorial staff doing most of the work without compensation [17]. Having limited resources - and limited technical and publishing expertise - makes it harder for small journals and publishers to meet new requirements.

In this study we aim to answer the following questions:

1.  How many open access journals are potentially Plan S-compliant?
2.  How does compliance relate to publisher size, business model and subject fields?
3.  Why are the requirements of Plan S especially challenging for small, non-APC financed open-access publishers?

## 2. Materials and Methods

We base our analysis on DOAJ's journal level metadata. DOAJ is considered the authoritative database of open access journals of scholarly quality. The metadata information is published by DOAJ as a csv file, a new file version is published every 30 min (see https://doaj.org/faq#metadata). We downloaded our file on 12 December 2018, at 14:50 CET; there are 12,350 journals in our DOAJ metadata file. This file contains data reported by the journals upon application, but important information is vetted by DOAJ's corps of editors.

We assume that data were mostly correct at the time of deposit, but we expect that not all data have been updated since deposit: we have, for example, during our other work observed that the APC charged often differs from the information found in DOAJ. Apart from APC amounts, most other information is more stable. A lack of information in the metadata file on when information last was updated makes any attempt to assess the average age of the information difficult - but due to the re-accreditation process, metadata must be from a date after March 2014.

Another issue with DOAJ metadata is that some questions in the application form allow free text responses and some questions are ambiguous, resulting in some columns in the metadata containing very different values. We have to the best of our ability tried to clean the data before doing the analyses. This is further discussed in the following sections.

An evaluation of the data quality provided by DOAJ has not been possible due to time constraints. This is a limitation—but given the purpose of this study and the role DOAJ is given in the Plan S requirements, we consider it a minor weakness. It is reasonable to assume that Coalition S will have to rely on metadata in DOAJ or other similar services to certify Plan S-compliant journals.

Based on the limitations noted above, we expect that our data could give a more negative picture than what is reality, as some journals may have improved their situation re the criteria, without updating the journal information in DOAJ. We assume, however, that this will not alter the results significantly. A special case is journals that have changed their status re charging APC or not—this could be a problem especially for journals having no APC in an introductory phase, and then converting to charging APCs. Some journals have missing information in fields we want to analyse— this is generally more of a nuisance than a problem.

The implementation guidance on Plan S lists technical requirements and recommendations for compliant open access journals and platforms. These are again divided into three sections. Section 9.1 lists basic mandatory criteria including requirements on copyright, licensing and peer review. Section 9.2 lists mandatory quality criteria. In addition, Section 9.3 lists recommended criteria including use of persistent identifiers (PIDs) for authors (such as ORCID), funders, institutions and so on. It is also recommended to directly deposit publications to repositories and have accessible and standardized data on citations in accordance with the Initiative for Open Citations.

The basic mandatory criteria and the mandatory quality criteria for Plan S-compliant journals, platforms and other venues are listed in their entirety in Table 1 under the column "Plan S Criterion". We have identified 14 criteria, of which four—G (mirror journals), H (transparent costing and pricing), L (data/code link) and M (interoperable metadata)—cannot be analysed using DOAJ journal metadata, while two criteria, A (DOAJ registry) and C (open availability) are fulfilled by default by any journal found in DOAJ. Under "Column header in DOAJ metadata" we have provided the header to the column in the DOAJ metadata we have used to analyse each criterion. We are left with eight criteria we can analyse—not to perfection, but to a reasonable degree. The wording in the criteria is not always

clear and what fulfils them not always self-evident, however we still have enough information to be able to do a meaningful analysis.

**Table 1.** Summary of Plan S criteria and DOAJ journal metadata used in the analyses.

| | Plan S Criterion | Analysable from DOAJ Journal Metadata | Column Header in DOAJ Journal Metadata |
|---|---|---|---|
| A (DOAJ-registry) | The journal/platform must be registered in the Directory of Open Access Journals (DOAJ) or in the process of being registered. (Implementation Guidance section 9.1) | | N/A. OK for all journals in the DOAJ |
| B (license) | The journal/platform must enable authors to publish under a CC BY 4.0 license (alternatively CC BY-SA 4.0 or CC0). (Implementation Guidance section 9.1) | Yes | "Journal license" |
| C (open availability) | All scholarly content must be openly accessible (journal website or dedicated platform) and free to read and download immediately upon publication, without any kind of technical or other form of obstacles. (Implementation Guidance section 9.1) | | N/A. OK for all journals in the DOAJ |
| D (copyright retention) | The journal/platform must offer authors/institutions the option of full copyright retention without any restrictions, i.e., no copyright transfer or license to publish that strips the author of essential rights. (Implementation Guidance section 9.1) | Yes | "Author holds copyright without restrictions?" |
| E (review) | The journal/platform must have a solid system in place for review according to the standards within the relevant discipline, and according to the standards of the Committee on Publication Ethics (COPE). Details on this must be openly available through the website. (Implementation Guidance section 9.1) | Yes | "Review process" |
| F (waivers) | The journal/platform must provide automatic APC waivers for authors from low-income countries and discounts for authors from middle-income countries. (Implementation Guidance section 9.1) | Yes | "Journal waiver policy (for developing country authors etc)" |
| G (mirror journals) | The journal must not have a mirror/sister subscription journal with substantial overlap in editorial board to avoid business models charging for both access and publication. Such journals will de facto be considered hybrid journals (see 'Transformative Agreements' below). (Implementation Guidance section 9.1) | No | |
| H (transparent costing and pricing) | Transparent costing and pricing: information on the publishing costs and on any other factors impacting the publication fees (for example cross subsidizing) must be openly available on the journal website/publishing platform. This must include details on direct costs, indirect costs and potential surplus. (Implementation Guidance section 9.2) | No | |
| I (DOIs) | Use of DOIs as permanent identifiers (PIDs with versioning, for example in case of revisions). (Implementation Guidance section 9.2) | Yes | "Permanent article identifiers" |
| J (long-term preservation) | Deposition of content with a long-term digital preservation or archiving programme (such as CLOCKSS). (Implementation Guidance section 9.2) | Yes | "Digital archiving policy or program(s)" |

**Table 1.** *Cont.*

| | Plan S Criterion | Analysable from DOAJ Journal Metadata | Column Header in DOAJ Journal Metadata |
|---|---|---|---|
| K (machine-readable full-text format) | Availability of the full text (including supplementary text and data when applicable and feasible) in machine readable format (for example XML), allowing for seamless Text and Data Mining (TDM). (Implementation Guidance section 9.2) | Yes | "Full text formats" |
| L (data/code link) | Linking to underlying data, code, and so on available in external repositories. (Implementation Guidance section 9.2) | No | |
| M (interoperable metadata) | High quality article level metadata – including cited references – in standard interoperable format, under a CC0 public domain dedication. Metadata must include complete and reliable information on funding provided by cOAlition S funders. (Implementation Guidance section 9.2) | No | |
| N (embedded licensing info) | Machine readable information on the Open Access status and the license embedded in the article. (Implementation Guidance section 9.2) | Yes | "Machine-readable CC licensing information embedded or displayed in articles" |

As noted above, not all Plan S criteria can be discussed through information available in DOAJ metadata. Analysing the criteria using DOAJ metadata is also reliant on an interpretation of what the criteria entail. Even though many of the criteria are self-evident, some of them are not, and further information on how to comply with the criteria will be necessary. Consequently, we do not aim to provide a list of Plan S-compliant journals. Journals fulfilling all of the criteria we have analysed are at least potentially Plan S compliant, whereas journals that fail to comply with these criteria will, by all accounts, not be compliant.

## 3. Results

A main requirement in the implementation guidance is that open access journals must be registered in DOAJ to be compliant with Plan S. DOAJ is a directory that indexes and provides access to open access, peer-reviewed journals. To be included in DOAJ, journals must fulfil certain scholarly and technical quality criteria.

The technical criteria of Plan S are generally in line with technical industry standards and best practices within scholarly publishing. However, there are probably not many publishers that currently meet every criterion. DOAJ's Principles of Transparency and Best Practice in Scholarly Publishing cover some of these criteria [18]. Other criteria are covered by the DOAJ seal [19]. To be awarded the DOAJ seal journals have to comply with the following seven conditions:

- uses DOIs as permanent identifiers;
- provides DOAJ with article metadata;
- deposits content with a long-term digital preservation or archiving program;
- embeds machine-readable CC licensing information in articles;
- allows generous reuse and mixing of content, in accordance with a CC BY, CC BY-SA or CC BY-NC license;
- has a deposit policy registered with a deposit policy registry;
- allows the author to hold the copyright without restrictions.

The seal signals that journals adhere to high level of publishing standards and best practice but has nothing to do with the scholarly quality of the material published in the journals [20]. Currently

nearly 1400 journals have been awarded the seal [21]. Although a journal's inclusion in DOAJ will ensure that some of the mandatory criteria in the implementation guidance are covered, not even the journals awarded the DOAJ seal can by default be said to fulfil every criterion.

A summary of the criteria in the technical requirements of Plan S, together with an indication of which criteria can be analysed using DOAJ journal metadata, is shown in Table 1 above. Below we look at the compliance rate for each criterion.

Criterion B (license) (Table 2)

This criterion is met by 5831 journals. CC BY, CC BY-SA or public domain are accepted as OK in our analysis.

**Table 2.** License compliance.

| License | Number of Journals |
|---|---|
| CC BY | 5091 |
| CC BY-NC | 2247 |
| CC BY-NC-ND | 2743 |
| CC BY-NC-SA | 951 |
| CC BY-ND | 139 |
| CC BY-SA | 739 |
| Public domain | 1 |
| Publisher's own license | 409 |
| (No data available) | 30 |
| **Total** | **12,350** |

Criterion D (copyright retention)

Fifty-six journals have no information about this. In 6318 journals authors do not retain copyright, while they do retain copyright in 5976 journals.

Criterion E (review) (Table 3)

DOAJ metadata lists five different forms of review. We consider all but "editorial review" or "(No data available)" to be compliant with criterion E. That means 189 of 12,350 journals do not meet this criterion, 12,161 do.

**Table 3.** Peer review compliance.

| Review Process | Number of Journals |
|---|---|
| Blind peer review | 3494 |
| Double blind peer review | 6001 |
| Editorial review | 133 |
| Open peer review | 135 |
| Peer review | 2531 |
| (No data available) | 56 |
| **Total** | **12,350** |

Criterion F (waivers)

DOAJ metadata state whether the journal has a waiver policy but give no detail about what the policy is (there is a link to information in the individual journal, though). Due to the limited information

available in the DOAJ metadata, we will in our analysis assume that all journals having a waiver policy meet the Plan S criterion, but this is not necessarily true. To be Plan S-compliant, the journal must provide automatic APC waivers for authors from low-income and discounts for authors from middle-income countries—these are the most common types of waivers. Morrison et al. [22] found that 80% of APC journals in DOAJ provide waivers or discounts for authors from low- to medium-income countries, but the study does not state if these are automatic waivers. Other types of discounts are for example discounts based on contributions to the journal or institutional membership. In another study on fee waivers for open access journals, Lawson [23] reported that 69% of the largest open access and toll access publishers offered waivers. However, only 36% offered automatic waivers to authors from low- and middle-income countries. Thus, having a waiver policy does not guarantee that journals have a fee waiver to authors from low- and middle-income countries or that the waiver is automatic. Our assumption that all journals having a waiver policy meet the Plan S criterion will likely give a more positive picture than what is true.

10,047 journals do not have a waiver policy, only 2303 have one, but this is a criterion that is only applicable to APC journals. There are 3244 journals that charge an APC listed in DOAJ, 59 have no APC information. 1400 of the APC journals (or those with no APC information) do not have a waiver policy, 1903 have one.

A strange fact is that 400 journals that state they do not charge APCs still have a waiver policy. Our impression is that the majority of these are open access journals published by larger publishers that publish journals both with and without APCs, possibly also journals that have a zero APC in an introductory period—and that the waiver information is a general clause inherited by default. For analytical purposes it is not meaningful to credit non-APC journals for having a waiver policy, this could mess up our analysis, so we set the waiver policy to "No" for these purposes. Another problem is that not all journals that go from not charging APCs to charging APCs change their information in DOAJ accordingly.

Criterion I (DOIs)

The DOI information field in DOAJ metadata contains a lot of various texts, including such as "In the process of acquiring DOI" etc. We consider that all journals where this field has the text string "DOI" in it, meet the criterion. 7209 journals meet this criterion according to our analysis, 5141 do not.

Criterion J (Long-term preservation)

CLOCKSS is the only example mentioned of what meets the criterion. There are more services, and many journals use more than one. We are not certain which of the services mentioned actually meet the criterion but have decided to include users of the following services that we find in the "Digital archiving policy or program(s)" field as meeting the criterion: LOCKSS, CLOCKSS, PKP PLN, Portico, PMC, Europe PMC, and PMC Canada. If this is a correct assumption, we have 3627 journals meeting the criterion, 8723 not meeting it.

Among those not meeting this criterion, 2105 journals have a value in the column "Archiving: national library", another 565 in the column "Archiving: other". The former of these columns contains nearly 400 different values, the latter nearly 275. Common to them is that it is difficult to understand the actual service used and there is no information about what this implies, so we have concluded that these journals are not Plan S-compliant. Plan S will need to define what constitutes an acceptable service, and create a list of compliant services, so that editors and publishers know what services to use.

Our restrictive interpretation of this criterion probably underestimates the number of journals that are compliant. An alternative would be to assume all journals that have a value in one of the relevant columns to be compliant, this would also probably be wrong.

Criterion K (machine-readable full-text format)

Only XML is mentioned as an acceptable format. From what we understand, HTML is also a format that meets the requirements. We have included journals where the full-text information

field contains the strings "HTML" or "XML" as compliant. That gives us 4530 compliant and 7820 non-compliant journals. If only XML is compliant, the numbers change to 1470 compliant and 10,879 non-compliant journals.

Criterion N (embedded licensing info)

DOAJ metadata give us information about whether machine-readable CC license information is embedded in the text files. 6610 journals meet this criterion, 5740 do not.

### 3.1. Publisher Size

We have analysed publisher size (Table 4), measured as the number of journals a publisher publishes, as one background variable, because we believe this to influence the capacity and competence to fulfil Plan S requirements. The field "Publisher" in the metadata is used to identify the publisher. There is a number of problems with this, which is discussed in [13] and we refer the reader to the discussion there. Another way of looking at publisher size is by looking at the number of articles a publisher publishes. Such numbers are not easily available and we have chosen to use the number of journals published as the measure of publisher size.

**Table 4.** Publisher size statistics.

| Publisher Size | No of Publishers | No of Journals | Percentage of Publishers | Percentage of Journals |
|---|---|---|---|---|
| 1 | 4446 | 4446 | 80% | 36% |
| 2 | 522 | 1044 | 9% | 8% |
| 3 | 187 | 561 | 3% | 5% |
| 4 | 108 | 432 | 2% | 3% |
| 5 | 69 | 345 | 1% | 3% |
| 6–10 | 137 | 1029 | 2% | 8% |
| 11–20 | 70 | 1019 | 1% | 8% |
| 21–50 | 33 | 1002 | 1% | 8% |
| 51–100 | 4 | 293 | 0% | 2% |
| >100 | 10 | 2179 | 0% | 18% |
| | 5586 | 12,350 | 100% | 100% |

Compared to the findings in 2010 [13], the smallest publishers measured by the number of journals they publish represent a slightly smaller percentage of publishers (80% now and 88% then), and a much smaller percentage of the journals (36% now, 55% then). The larger publishers (50+ journals) have grown from 0.2% of publishers to 0.3, now representing 20% of journals compared to 12.9% then. Larger publishers have become more important to the total volumes published, smaller publishers less important. These findings are largely confirmed by Morrison [14]. Using a larger dataset containing over 14,000 open access journals, she finds that 82% of publishers publish one journal, while only 2% publish 10 journals or more. There is also a decrease in the total number of journals published by smaller publishers from 47% in 2014 to 37% in 2017.

The 10 largest publishers are Dove Medical Press (102 journals), Taylor & Francis Group (138 journals), SAGE Publishing (149), Wolters Kluwer Medknow Publications (177), MDPI AG (181), SpringerOpen (197), Hindawi Limited (250), BMC (321), Sciendo (326) and Elsevier (338), a total of 2179 journals or 17.6% of the total number of journals in DOAJ.

SpringerOpen and BMC are both parts of SpringerNature, in addition the Nature Publishing Group has 47 journals, bringing SpringerNature to the top of the list with a total of 565 journals. Sciendo is a publishing service started by De Gruyter, who has 56 journals under their own brand,

making Sciendo + De Gruyter with 382 journals the second largest publisher after SpringerNature, with Elsevier in third place. Adding these smaller brands brings the total up to 2282 journals, 18.5% of the total in DOAJ.

### 3.2. A More General Picture

To get a clearer picture, we have given compliance of a criterion the value 1, non-compliance 0. If we then sum these values to a total score, we see to which extent a journal meets all eight criteria we analyse—8 being a perfect score, i.e., compliant on all criteria analysed, 0 being non-compliant on all criteria. For journals not charging APC, 7 is the perfect score as a waiver policy is meaningless for these journals.

We do suspect that there is a correlation between scores and

a.　　Charging an APC or not
b.　　The size of the publisher as measured in the number of journals the publisher publishes

This is a very general assumption, and a number of journals will show this not to be true on the individual journal level. However, some of these criteria cannot be met without financial resources or good publishing competence. Charging an APC gives you a chance to meet financial needs, and publishing many journals enables you to develop publishing competence [13,24]. Although there are many business models in open access publishing, there is a clear tendency that the larger publisher journals are financed by APCs.

If we look at the total score and how that is distributed depending on whether the journal has an APC or not, we get the picture shown in Table 5.

**Table 5.** Sum of compliance and APC status.

| APC | Sum of Criteria Fulfilled (Number of Journals) | | | | | | | | | | |
|---|---|---|---|---|---|---|---|---|---|---|---|
| | 0 | 1 | 2 | 3 | 4 | 5 | 6 | 7 | 8 | Total | Avg. score |
| No[1] | 54 | 876 | 2029 | 2408 | 1754 | 1398 | 332 | 255 | N/A | 9106 | 3.29 |
| Yes | 0 | 76 | 225 | 380 | 347 | 349 | 792 | 245 | 830 | 3244 | 5.52 |
| **Total** | **54** | **952** | **2254** | **2788** | **2101** | **1747** | **1124** | **500** | **830** | **12,350** | **3.34** |
| APC | Sum of Criteria Fulfilled (Percentage) | | | | | | | | | | |
| | 0 | 1 | 2 | 3 | 4 | 5 | 6 | 7 | 8 | Total | |
| No | 1% | 10% | 22% | 26% | 19% | 15% | 4% | 3% | N/A | 100% | |
| Yes | 0% | 2% | 7% | 12% | 11% | 11% | 24% | 8% | 26% | 100% | |
| **Total** | **0%** | | | | | | | | | **100%** | |

The upper part is in absolute numbers, the lower in percentages—each line sums up to 100%.

Only 1085 (8.8%) of the journals registered in DOAJ meet all criteria. Of the journals 255 are non-APC journals (score 7) and 830 are APC journals (score 8). 59% of non-APC journals meet three or fewer criteria (less than half of the criteria), while 32% of APC journals meet four or fewer criteria (half or less of the criteria). So there is a marked tendency for APC journals to meet more criteria than non-APC journals. This is also reflected in the average number of criteria met, 3.29 for non-APC journals and 5.52 for APC journals. The difference of 2.23 is larger than the one extra criterion (waiver) would necessitate.

---

[1]　　In the data, there are 59 journals where there is no information on whether they charge an APC or not. We have included this in the "No" category, as the majority of journals belong here, and we assume that journals charging an APC will be less likely to lack information about this.

To ensure we are comparing the same for all journals, we have here excluded criterion F (Waivers) from the analysis. We see from the above (Table 6) that smaller publishers have a larger percentage of their journals in the left part of the table, and fewer in the right, while the larger publishers have few in the left and many in the right part of the table. This means there is a correlation between the publisher size and the ability to comply with the criteria. We see from the averages, however, that this effect really only shows for publishers having 20 or more journals. Excluding the values for Criterion F (Waivers) we find a correlation of 0.525 between publisher size (not grouped into size intervals) and the sum score on the seven criteria. This could indicate that there is some "critical mass" needed for size to influence the scores, and this point lies somewhere above 10 journals.

**Table 6.** Publisher size and sum of criteria compliance.

| No of Journals | Sum of Criteria Fulfilled (Number of Journals) | | | | | | | | | |
|---|---|---|---|---|---|---|---|---|---|---|
| Publisher size | 0 | 1 | 2 | 3 | 4 | 5 | 6 | 7 | Total | Avg. score |
| 1 | 23 | 505 | 1063 | 1276 | 900 | 508 | 136 | 35 | 4446 | 3.1 |
| 2 | 15 | 108 | 246 | 295 | 213 | 121 | 33 | 13 | 1044 | 3.1 |
| 3 | 4 | 63 | 140 | 153 | 110 | 62 | 21 | 8 | 561 | 3.1 |
| 4 | | 39 | 112 | 112 | 81 | 41 | 36 | 11 | 432 | 3.3 |
| 5 | | 17 | 88 | 105 | 75 | 42 | 13 | 5 | 345 | 3.3 |
| 6–10 | 3 | 89 | 248 | 319 | 189 | 131 | 30 | 20 | 1029 | 3.2 |
| 11–20 | 4 | 101 | 225 | 230 | 166 | 155 | 69 | 69 | 1019 | 3.5 |
| 21–50 | 4 | 23 | 144 | 256 | 165 | 151 | 104 | 155 | 1002 | 4.2 |
| 51–100 | | | 18 | 72 | 33 | 110 | 26 | 34 | 293 | 4.5 |
| >100 | 1 | 18 | 1 | 27 | 212 | 909 | 185 | 826 | 2179 | 5.7 |
| **Total** | **54** | **963** | **2285** | **2845** | **2144** | **2230** | **653** | **1176** | **12,350** | **3.7** |
| **No of Journals** | **Sum of Criteria Fulfilled (Percentage)** | | | | | | | | | |
| Publisher size | 0 | 1 | 2 | 3 | 4 | 5 | 6 | 7 | Total | |
| 1 | 1% | 11% | 24% | 29% | 20% | 11% | 3% | 1% | 100% | |
| 2 | 1% | 10% | 24% | 28% | 20% | 12% | 3% | 1% | 100% | |
| 3 | 1% | 11% | 25% | 27% | 20% | 11% | 4% | 1% | 100% | |
| 4 | | 9% | 26% | 26% | 19% | 9% | 8% | 3% | 100% | |
| 5 | | 5% | 26% | 30% | 22% | 12% | 4% | 1% | 100% | |
| 6–10 | 0% | 9% | 24% | 31% | 18% | 13% | 3% | 2% | 100% | |
| 11–20 | 0% | 10% | 22% | 23% | 16% | 15% | 7% | 7% | 100% | |
| 21–50 | 0% | 2% | 14% | 26% | 16% | 15% | 10% | 15% | 100% | |
| 51–100 | | | 6% | 25% | 11% | 38% | 9% | 12% | 100% | |
| >100 | 0% | 1% | 0% | 1% | 10% | 42% | 8% | 38% | 100% | |
| **Total** | **0%** | **8%** | **19%** | **23%** | **17%** | **18%** | **5%** | **10%** | **100%** | |

*3.3. HSS/STEM Journals*

Based on information about "Subjects" in the DOAJ data, we have grouped journals into HSS and STEM journals, except for 14 journals where this information is missing. The field "Subjects" in the metadata file contains information about scholarly field. Alternative classifications are separated by "|", while higher level and lower level subject classifications are separated by ":". We have assumed the first classification to be the most relevant in case of alternative classifications and have used the high

level term (before the first ":") as a classification of subject field. This left us with 20 subjects, which we manually have sorted into HSS and STEM.

Combining this with information on whether journals charge APC or not (excluding 59 where this information is not available and the 14 where subject is lacking) and publisher size in terms of journals published, we get this overview (Table 7).

**Table 7.** APC and non-APC journals in HSS and STEM sorted by publisher size.

| APC | | HSS/STEM | | |
|---|---|---|---|---|
| | **Publisher Size Group** | **HSS** | **STEM** | **Total** |
| No | 1 | 2396 | 1399 | 3795 |
| | 2 | 551 | 327 | 878 |
| | 3 | 297 | 156 | 453 |
| | 4 | 200 | 118 | 318 |
| | 5 | 170 | 101 | 271 |
| | 6–10 | 632 | 239 | 871 |
| | 11–20 | 599 | 259 | 858 |
| | 21–50 | 457 | 226 | 683 |
| | 51–100 | 77 | 50 | 127 |
| | >100 | 210 | 574 | 784 |
| **No Total** | | **5589** | **3449** | **9038** |
| Yes | 1 | 218 | 416 | 634 |
| | 2 | 52 | 99 | 151 |
| | 3 | 43 | 61 | 104 |
| | 4 | 43 | 70 | 113 |
| | 5 | 39 | 35 | 74 |
| | 6–10 | 61 | 95 | 156 |
| | 11–20 | 35 | 121 | 156 |
| | 21–50 | 75 | 238 | 313 |
| | 51–100 | 19 | 146 | 165 |
| | >100 | 116 | 1257 | 1373 |
| **Yes Total** | | **701** | **2538** | **3239** |
| **Grand total** | | **6290** | **5987** | **12,277** |

We see that a majority of journals in DOAJ are non-APC journals, and the majority of them are HSS journals from small publishers. Publishers with one or two journals have the majority of non-APC HSS journals, the situation is nearly the same for non-APC STEM journals. Among the APC journals, the vast majority are among STEM journals. And the larger publishers publish a large part of the journals, especially in STEM. These results are aligned with Crawford's [15]. Thus, we are looking at a world characterized by many small HSS publishers publishing without charging APC, and fewer and larger STEM publishers financing activities by charging APCs.

*3.4. Meeting the Criteria*

The 14 different criteria we have identified differ significantly, in what is needed to comply. Some criteria can be met by making the right policy decisions; some need competence and some degree of funding to enable journals to meet them. Criterion A (DOAJ registry) and C (open availability) can

be said to be fulfilled by default by being registered in DOAJ, hence we exclude them from our analyses. Meeting criterion B (license), D (copyright retention), E (review), F (waivers), G (mirror journals), and H (transparent costing and pricing) is mostly a question of policy. Criterion I (DOIs), J (long-term preservation), K (machine-readable full-text format), L (data/code link), M (interoperable metadata) and N (embedded licensing info) are reliant on available technical infrastructure and technical competence, or funding to buy external services.

We can group the criteria we can analyse through the metadata into policy requirements and technical requirements. The policy requirements consist of criterion B (license), D (copyright retention), E (review) and F (waivers). The technical requirements consist of criterion I (DOIs), J (long-term preservation), K (machine-readable full-text format) and N (embedded licensing info).

If we look at policy criteria (Table 8), we find that there is not much difference between APC journals and non-APC journals. Remember: non-APC journals only have three policy criteria to comply with, APC journals four, including waiver policy. Averages indicate that the only "effect" is that the largest APC charging publishers seem to be somewhat more compliant than the smaller ones, this effect is not to be found among the publishers not charging APC.

**Table 8.** Policy criteria compliance and publisher size.

| Number of Journals | Number of Policy Criteria Met | | | | | | |
|---|---|---|---|---|---|---|---|
| APC or Not/Publisher Size | 0 | 1 | 2 | 3 | 4 | Total | Avg. Number of Criteria Met |
| **No** | | | | | | | |
| 1 | 30 | 1296 | 1516 | 958 | N/A | 3800 | 1.9 |
| 2 | 6 | 282 | 392 | 199 | N/A | 879 | 1.9 |
| 3 | 1 | 160 | 189 | 103 | N/A | 453 | 1.9 |
| 4 | 1 | 92 | 156 | 69 | N/A | 318 | 1.9 |
| 5 | | 63 | 120 | 88 | N/A | 271 | 2.1 |
| 6–10 | 3 | 292 | 382 | 195 | N/A | 872 | 1.9 |
| 11–20 | 2 | 288 | 308 | 262 | N/A | 860 | 2.0 |
| 21–50 | 2 | 204 | 331 | 146 | N/A | 683 | 1.9 |
| 51–100 | 1 | 15 | 109 | 2 | N/A | 127 | 1.9 |
| >100 | | 430 | 174 | 180 | N/A | 784 | 1.7 |
| **No total** | **46** | **3122** | **3677** | **2202** | **N/A** | **9047** | **1.9** |
| **Yes** | | | | | | | |
| 1 | 1 | 144 | 245 | 167 | 77 | 634 | 2.3 |
| 2 | | 36 | 49 | 50 | 16 | 151 | 2.3 |
| 3 | | 22 | 42 | 26 | 14 | 104 | 2.3 |
| 4 | | 16 | 32 | 49 | 17 | 114 | 2.6 |
| 5 | | 15 | 27 | 16 | 16 | 74 | 2.4 |
| 6–10 | | 40 | 61 | 34 | 21 | 156 | 2.2 |
| 11–20 | | 46 | 73 | 21 | 16 | 156 | 2.0 |
| 21–50 | | 24 | 99 | 62 | 130 | 315 | 2.9 |
| 51–100 | | | 50 | 37 | 79 | 166 | 3.2 |
| >100 | | 57 | 509 | 171 | 637 | 1374 | 3.0 |
| **Yes total** | **1** | **400** | **1187** | **633** | **1023** | **3244** | **2.7** |
| **Grand total** | **47** | **3522** | **4864** | **2835** | **1023** | **12,291** | **2.1** |

If we look at the technical criteria, we get a different picture (Table 9).

**Table 9.** Technical criteria compliance and publisher size.

| Number of Journals | Number of Technical Criteria Met | | | | | | |
|---|---|---|---|---|---|---|---|
| **APC or Not/Publisher Size** | **0** | **1** | **2** | **3** | **4** | **Total** | **Average Number of Criteria Met** |
| **No** | | | | | | | |
| 1 | 1214 | 1362 | 823 | 350 | 51 | 3800 | 1.1 |
| 2 | 268 | 319 | 199 | 79 | 14 | 879 | 1.1 |
| 3 | 147 | 166 | 87 | 38 | 15 | 453 | 1.1 |
| 4 | 111 | 113 | 73 | 16 | 5 | 318 | 1.0 |
| 5 | 90 | 110 | 37 | 30 | 4 | 271 | 1.1 |
| 6–10 | 228 | 348 | 219 | 47 | 30 | 872 | 1.2 |
| 11–20 | 227 | 261 | 200 | 99 | 73 | 860 | 1.5 |
| 21–50 | 97 | 243 | 157 | 120 | 66 | 683 | 1.7 |
| 51–100 | 17 | 65 | 6 | 32 | 7 | 127 | 1.6 |
| >100 | 1 | 1 | 34 | 316 | 432 | 784 | 3.5 |
| **No total** | **2400** | **2988** | **1835** | **1127** | **697** | **9047** | **1.4** |
| **Yes** | | | | | | | |
| 1 | 152 | 214 | 131 | 93 | 44 | 634 | 1.5 |
| 2 | 30 | 38 | 44 | 22 | 17 | 151 | 1.7 |
| 3 | 17 | 42 | 22 | 12 | 11 | 104 | 1.6 |
| 4 | 21 | 36 | 8 | 22 | 27 | 114 | 2.0 |
| 5 | 14 | 20 | 27 | 7 | 6 | 74 | 1.6 |
| 6–10 | 23 | 56 | 35 | 18 | 24 | 156 | 1.8 |
| 11–20 | 20 | 32 | 22 | 22 | 60 | 156 | 2.4 |
| 21–50 | 9 | 48 | 25 | 36 | 197 | 315 | 3.2 |
| 51–100 | 3 | 4 | 48 | 36 | 75 | 166 | 3.1 |
| >100 | | | 11 | 195 | 1168 | 1374 | 3.8 |
| **Yes total** | **289** | **490** | **373** | **463** | **1629** | **3244** | **2.8** |
| **Grand total** | **2689** | **3478** | **2208** | **1590** | **2326** | **12,291** | **1.8** |

We see that most non-APC journals satisfy only a few of the technical criteria (average of 1.4), while a majority of APC journals (1629 of 3244) satisfy them all. The average APC journal satisfies 2.8 criteria, double the score of the non-APC journals. The level of compliance increases with publisher size, for both APC and non-APC journals, especially when the publisher size comes above 10 journals. Correlation analyses shows a 0.602 correlation between publisher size and score on the technical criteria, 0.5 and 0.599 for non-APC and APC journals respectively.

Looking briefly at the four criteria, we are not able to analyse using DOAJ metadata. We find that two criteria (G and H) are only applicable to fee-charging journals. Criterion G states that journals must not have a mirror/sister subscription journal with substantial overlap in editorial board. This criterion can be seen as a pre-emptive measure as we do not expect that there are many existing journals that fall into this category. Unlike traditional hybrid journals that publish both closed and open content, there are no criteria today that prevent open access journals with mirror/sister subscription journals from inclusion in DOAJ. It might be more difficult to comply with the requirement on openly available information on costing and pricing impacting the publication fees (Criterion H). Information on price is already provided by APC journals. Providing information on publishing cost is a more complicated matter [12]. Commercial publishers will see this information as trade secrets and most expense are overheads, they are infrastructural and competence costs not easily allocated to individual articles. Linking to underlying data, code and so on (criterion L) from published articles is becoming more common, but studies show there has been a weak adoption of data policies in open access journals [25].

What Criterion M entails is not self-evident requiring high quality metadata, including cited references under a public domain dedication. If we interpret it correctly, it will be met by publishers participating in the Initiative for Open Citations I4OC (https://i4oc.org/) which is also a recommendation in the implementation guidance on Plan S. Many publishers are already participating and the portion of publications with open citations is now over 50%. However, reference editing and tagging is not an easy operation, especially if the technical infrastructure and competence is missing [12]. Using the Crossref Funder Registry (https://www.crossref.org/services/funder-registry/) will likely be sufficient to comply with the second part of criterion M requiring complete and reliable information on funding.

## 4. Discussion

In the pioneering years of open access (the 1990s), the majority of open access journals were operated on a voluntary basis by small groups of scholars and published on technically simple platforms provided by the editors' institutions [16,26]. The typical journal was a stand-alone journal, publishing few articles and operating without a budget with limited or no direct expenses or out-of-pocket cost. Later, new business models emerged, making it possible to run professional and commercial open access publishing on a larger scale. Since around 2003, the open access market has become increasingly dominated by professionally published journals, often financed by charging APCs [26]. Now, the majority of large publishers run open access journals or publish hybrid open access in their subscription journals [24].

The cost of publishing has been much debated. The estimation of cost provides very different results depending on publisher and journal, ranging from over $4000 to under $10 per article [27]. Conducting a survey in 2009 of journals using Open Journal Systems, Edgar and Willinsky identified the vast majority of journals in their sample as independent or scholar-publisher titles [17]. The journals had very active editors with many participating in editing, proofreading and layout tasks mostly without remuneration. 44% of the journals reported having zero revenue and 29% were operating with zero expenses. The average cost per article in the study was estimated to $188.

The open access scene has changed considerably since the 1990s, but still the vast majority of open access publishers are small, only publishing a single journal, largely stemming from universities [14,15]. An often-mentioned challenge for the small scholar-published open access journals is the question of scalability [26]. Will the publishing and business model support a growth in published articles? Implied in this question is a relation between size and sustainability of a journal. However, viewing journal publishing as part of a community building process around a new field of inquiry [28] provides a different reasoning to the publishing purpose and basis for existence. A study by Björk, Shen and Laakso on independent scholar-published open access journals founded prior to 2002, showed that 51% of the journals were still active in 2014 and that only 8% had started charging APC [26]. Although the median number of published articles per year had risen from 11 to 18, the study concludes that even the successful independent journals tend to be rather small.

There are challenges for small publishers related to economy, efficiency and expertise. A study by Morrison, interviewing 15 persons involved in producing small scholar-led journals showed that sustainable funding was the major concern [29]. Björk, Shen and Laakso mention the dependency on individual persons along with uncertainty concerning long-time support as dangers [26]. Conducting an analysis on the size distribution of open access publishers in 2010, Frantsvåg [13] investigated how many journals deliver article metadata to DOAJ and found that less than half of the registered journals did. Having the article metadata in DOAJ is an efficient method for disseminating the research and free and fairly easy to do. However, many small publishers did not use this functionality. In 2014 DOAJ introduced new criteria for inclusion and required journals that were already registered to reapply before April 2016, in order to be kept in the registry. Looking at journals that were removed from DOAJ after April 2016, Frantsvåg [30] found that the single journal publishers lost nearly one-third of journals while the percentage of lost journals steadily decreased in relation to publisher size. He concluded that the removal of journals has little to do with scholarly quality but is primarily a result of failure

to reapply. This leads to the understanding that many small publishers seem to have lacked the competence or resources necessary to understand or go through the re-application process. Our data points to a correlation between size and the ability to comply with the Plan S criteria. The correlation is strongest for publishers with a "critical mass" of more than 10 journals and could indicate that co-operation between journals, publishers and/or institutions would be beneficial to smaller publishers taking advantage of the economies of scale in scholarly publishing [31].

To illustrate the challenges of the Plan S criteria for publishers and journals, we will use Open Journal Systems (OJS) (https://pkp.sfu.ca/ojs/) as a case. OJS is a journal management and publishing system developed by The Public Knowledge Project (PKP) with the purpose of enhancing the quality of scholarly communication and global knowledge sharing. OJS is an open source software and is freely available, making it the system of choice for small and non-commercial publishers. In 2017 nearly 10,000 journals were reported to be using OJS [32]. Journals using OJS are distinguished from other journals by the number of journals that are published by scholars, the share of journals from developing countries and low operating budgets [17].

In our data, we find that almost 5000 journals listed in DOAJ state that they use OJS as publishing platform, of which the vast majority are non-APC journals. OJS offers functionality to comply with some of the criteria. For instance DOIs can be automatically assigned to articles and metadata delivered to Crossref (criterion I). PKP has also established a long-term digital preservation program (PKP PLN) which is integrated in OJS offering journals free long-term archiving (criterion J). In a future release OJS will also supply citations data to Crossref and thus comply (at least partially) with criterion M (interoperable metadata). The main challenge is the availability of full text in machine-readable format (criterion K). Most journals publish only PDFs, which are not machine readable, and the standard workflow is based on converting Word files to PDFs. For a number of years, PKP has been working on an XML conversion tool alongside an XML editor (https://pkp.sfu.ca/open-typesetting-stack/). These services are however still in beta status and are not to be exclusively relied upon for production. Using these services today would require technical competence and time-investment. The same applies if one was to generate XML or HTML files outside OJS.

Although services for DOI and digital preservation are integrated in OJS, we see from our DOAJ data that these are not widely used. Under half of the OJS journals use DOIs and even fewer have a digital preservation policy or program. A reason for not using DOIs could be that there is a cost linked with it as noted in a survey conducted by DOAJ [33]. But failing to use a digital preservation service that is free and easy to use, is probably more a question of missing publishing and technical competence. Failing to meet these basic requirements, which are easily solvable through the publishing system, illustrates the difficulty many small publishers will have complying with the more challenging technical requirements of Plan S.

One could argue that running a journal should require a certain competence and that scholars would be better served not publishing in journals failing to meet basic technical standards or missing basic technical or publishing competence. However, it is important to note that these technical requirements are not related to the scholarly quality of the published content. Only one of the Plan S criteria addresses scholarly quality. Criterion E says that journals/platforms must have a solid system in place for review according to standards within the relevant discipline and COPE. As we have already established, almost all journals registered in DOAJ meet this criterion.

The small open access journals and publishers have played a major part in promoting open access and making research results openly available in the last 25 years. cOAlition S recognises this contribution by explicitly acknowledging "the importance of a diversity of models and non-APC based outlets" [4]. The implementation guidance on Plan S also states that cOAlition S "does not favour any specific business model for Open Access publishing or advocate any particular route to Open Access given that there should be room for new innovative publishing models". However, as we see in our study, the requirements clearly favours the large over the small publishers and the well-funded, mainly APC financed, over the low-funded journals.

## 5. Limitations

The results in this study are based on matching DOAJ metadata with the Plan S criteria. As discussed above, there is no perfect match between metadata and criteria and there are also uncertainties on how some of the criteria are to be understood. This is especially relevant for criterion F (waivers) and criterion J (long-term preservation), but also to some extent criterion K (machine-readable full-text format). For criteria F (waivers) and K (machine-readable full-text format) we may have accepted too many journals as compliant, while for criterion J (long-term preservation) too few. This might affect the results, but not our overall conclusion that a majority of journals listed in DOAJ are not currently compliant with Plan S.

This study focuses on Plan S compliance of open access journals. The study does not cover the challenges faced by open access repositories to be Plan S-compliant, nor the ways in which researchers can make their scholarly articles Plan S-compliant. However, it should be noted that it will still be possible for authors to publish in non-compliant journals, as long as they are granted permission to make a peer-reviewed version immediately available, under CC BY, CC BY-SA or CC0 in an open access repository. This option would be available as default by publishing in open access journals under one of these licenses, and as such be a viable option for open access journals that are unable to meet the technical requirements. Nevertheless, depositing a copy in a repository puts additional work on authors and the journals will probably benefit from being available on a list of Plan S-compliant journals both in terms of visibility and as a sign of quality.

## 6. Conclusions

The goal of Plan S is full and immediate open access to publications from publicly funded research. To achieve this there must be available publishing venues that are aligned with this goal. There has been much debate among researchers on the consequences Plan S might have in limiting their opportunity to publish in traditional (and prestigious) toll access journals. As evident in this study the requirements in the implementation guidance on Plan S might also have an adverse effect on available open access journals.

Limiting our study to the criteria we can analyse using DOAJ metadata, we find that 8.8% (1085 of 12,350) of open access journals meet all of these criteria. Fulfilment of the remaining criteria might result in even fewer Plan S-compliant open access journals. Furthermore, there is a clear relation between journals charging APC, publisher size and the ability to comply with the criteria. Only 2.8% of non-APC journals and 25.6% of APC journals meet all criteria according to our analysis. Looking at academic disciplines it is clear that the humanities and social sciences will be most affected since the open access journals in these segments are usually smaller and free to publish in.

We are not arguing that these requirements should not be made. But we want to warn that the current timeline will pose a threat to a number of open access journals of good scholarly quality that scholars do not want to lose. The current timeline will remove the non-APC journals from the market, leaving APC journals the winners.

*Some Recommendations to cOAlititon S*

- Invest in technical infrastructure that will enable journals to meet the technical requirements. The tools must be freely available, open source and not require a high level of technical competence to use. For instance, the publishing system OJS is currently used by almost 5000 journals in DOAJ. Supporting the development of OJS to be able to deliver on all of the requirements would be an efficient and inexpensive way to enable many journals to be Plan S-compliant.
- Consider the possibility of different technical requirements for APC and non APC journals, or at least different time frames for implementation of the requirements, possibly with different dates for different requirements. As it is still unclear what some of the requirements imply, even competent

publishers may not know how to position themselves to be Plan S-compliant. For the smaller publishers, the current time frame is impossible to comply with.

- Plan S need to find or develop, and finance, services that can certify Plan S-compliant journals. In order to do that, one needs to define what are acceptable responses to the requirements about text format and archiving. The Plan S certification service will also need to certify the archiving services and define the acceptable text formats, in order to be able to certify Plan S-compliant journals.

**Supplementary Materials:** The data of this article will be made openly available at DataverseNO, https://doi.org/10.18710/YBGGTV. Detailed descriptions of the data will be available, as part of the data deposit.

**Author Contributions:** Conceptualization, T.S. and J.E.F.; methodology, J.E.F.; formal analysis, J.E.F.; writing—original draft preparation, T.S. and J.E.F.; writing—review and editing, T.S. and J.E.F.

**Funding:** This research received no external funding.

**Acknowledgments:** Thanks to our colleagues Aysa Ekanger and Ellen Nierenberg, both from UiT The Arctic University of Norway, for commenting on both content and language. Errors and omissions of fact and of grammar and spelling remaining are of our own making.

**Conflicts of Interest:** Jan Erik Frantsvåg is a member of MDPI Publications editorial board and the DOAJ advisory board. He is also engaged in the University Library publishing service, Septentrio Academic Publishing, which publishes non-APC open access journals. Tormod Eismann Strømme is engaged in the University Library publishing service, Bergen Open Access Publishing, which publishes non-APC open access journals.

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
