# Peer review of "Few Open Access Journals Are Compliant with Plan S"

_publications, doi:10.3390/publications7020026_

Reviewer 1 Report

Brace yourself before opening the attachment. This is a rigorous critique that is merited by your work in both positive and negative senses (it needs a lot of work, but what you are doing is important and worth it). 

Author Response

Please see attached file where we have both general comments and replies to individual comments by the reviewers.

Reviewer 2 Report

This is a helpful and important paper, and illustrates a good use of DOAJ data. I also appreciate the recommendations at the end, especially about investing in infrastructure that can make compliance with Plan S a reality. For example, automatic XML conversion from PDF or RTF files would make a wonderful investment that would help many journals adhere to the requirements. 

This work is so important at this juncture that I want this paper to be the best it can be. I have a few points where some clarification or a little bit of additional information is needed to help readers more easily understand the document and to give some of your arguments/discussion a bit of context, and two requested revisions, one of which can be done in either a major or minor way, and the other is necessary. 

First, clarifications/context:

1. This is an easy one - giving letters to the criteria gets confusing later in the paper. It would be better to give each criteria a short name, e.g. Criterion DOI or Criterion Copyright. That would make especially the last few pages much easier to read so readers don't have to flip back to remember what Criteron K is! 

2. Your first paragraph of the Discussion section (starting line 310) and in the Conclusion between lines 341 and 347 is very helpful about understanding the relationship between your DOAJ dataset and the discussion about APCs, funding, infrastructure, and ability to comply with the requirements. In particular, the information about the number of publishers who did not re-apply when the DOAJ Seal criteria were introduced and how you view that information as relevant to the broader discussion. I believe this information should come earlier in the paper, perhaps even in the introduction section, to underscore what we are reading about in the numbers. 

3. I think you can word your ideas about infrastructure cost and the privileging of costly infrastructure in Plan S even more strongly - drawing a more direct line between lack of APC funding to the inability to afford technology or infrastructure that is required for Plan S compliance. You might want to reference some of the costs of publishing OA here to further make this point. You can also connect this to the difference between HSS and STEM disciplines - much of that infrastructure privileges STEM disciplines and was made for it. This will also help the strength of your recommendations at the end. 
I know it is bad form for a reviewer to recommend a paper that she was a co-author of, but some of the relevant literature regarding the costs of publishing OA and the infrastructure costs were summarized in a 2016 paper by Tennant et al., in the section "The economic impact of Open Access" - see here: https://www.ncbi.nlm.nih.gov/pmc/articles/PMC4837983/#__sec7title 
You can definitely use some of those references if you are not familiar with them already - and of course anything newer! I would love to see that additional context of the costs of publishing literature make it into this discussion. 

Here are the two most major revisions:

1. Throughout the discussion of APCs, it might be worthwhile here to mention one, two, or some of the other business models for OA beyond APCs and make a comparison, especially if you believe these other models have a large presence in your "non-APC charging journals" data. Here's a list of OA business models: http://oad.simmons.edu/oadwiki/OA_journal_business_models 

And here's a paper published in 2016 that may be a helpful resource for this as well: https://dash.harvard.edu/handle/1/27803834 

For example, you could look at the "institutional subsidies" category and discuss how that model would interact with Plan S requirements to show more strongly how Plan S favors the APC model. 

This suggestion is not required to implement, although the other business models and a reference to them should be included as at least a brief mention. 
2. Throughout the paper, you often present in your data tables numbers and percentages from two (or more) groups of journals that have quite different sizes. Sometimes percentages are provided, and sometimes raw numbers. For example, tables 8 and 9 give raw numbers of journals, when the different total populations for each category (APC yes or no) are quite different (thousands of journals) - one group has 9047 journals and the other 3244. This makes it difficult to understand the scope of what we are seeing, or if the differences are significant. Earlier in the paper, you give percentages, which help a little bit. I think a more robust statistical test here is warranted. You need to conduct some statistical analysis here, e.g. using a two-sample t-test, to see if the behaviors between the two unequal size groups are truly similar or different. I am not a statistics expert and I'm not sure this particular test is the one you want for this group, but this paper really needs information on whether the behavior is significantly different between the many ways that you slice and group this data set. It is hard to tell from percentages and raw numbers alone, and this additional level of understanding about statistical significance will take this paper to the level it needs to be for publication.

I thank the authors for their work on this important issue and look forward to seeing this paper in publication. I hope my suggestions are useful in refining the article. 

Author Response

(The authors gave the same response as above.)

Reviewer 3 Report

I am pleased to recommend publication of this article after minor revisions. This is a timely topic, since feedback on Plan S requirements is being sought right now, and this article addresses important issues with it. The description and understanding of Plan S in the article is accurate. However, the language needs some copyediting throughout. I’ve not addressed all the necessary changes around grammar, phrasing etc. in my specific points listed below – the article needs a thorough read-through. I hope this won’t delay publication too much but I think it is necessary.

DOAJ as a data source is appropriate, given it’s role in the current Plan S requirements. To the best of knowledge the methodology is sound. For some of the analysis, such as on criterion F (fee waivers), the available data is not robust enough to satisfy the funders if they were monitoring policy compliance but is good enough for the preliminary analysis provided by this article. As stated in lines 211-13, Plan S funders still need to define their requirements a lot more precisely (and fund the infrastructure to allow this!).

It’s great that the authors have provided a supplementary data file with so much of the analysis in it. Additional description of the data – either in a tab on the spreadsheet, or a separate file – would make it even more useful because it’s not clear to me what every tab means.

Lines 110-12 state: ‘For this analysis, we assume all criteria that we do not have data about, to be fulfilled. This is of course a major weakness, but any error here will not make the overall situation look better.’ However, I don’t think it’s safe to assume these criteria fulfilled, in fact the opposite is more likely (especially criterion H). It’s good that the weakness is acknowledged, but returning to this point later in the discussion section would reinforce the amount of work still to be done if journals are to become compliant.

The conclusion of the discussion section (lines 328-332) – and the Conclusion section itself – follow logically from the preceding analysis, and make extremely important points for Plan S funders to understand. The recommendation that Plan S funders leverage OJS development to meet technical requirements is exactly what I intend to recommend to the consultation feedback myself.

Minor individual points:

- p.1 line 27, I would replace ‘scientists’ with ‘researchers’.

- p.1 line 32, change ‘common domain’ to ‘public domain’, because ‘public domain’ is a precise legal phrase that more accurately reflects the CC0 license.

- p.1 line 44, signed by more than 1500 what? Add word after ‘1500’ (e.g. ‘people’, ‘researchers’). Or rephrase the sentence.

- p.1 line 45, change ‘An other’ to ‘Another’

- p.3 line 94, please state the year of comparison (2010) in the text.

- p.3 line 121, I think you mean Table 2?

- p.4 Table 2 row A, I would add a dash or ‘n/a’ to column 3 rather than leaving it blank, to make it clear that it is intentionally empty.

- pp.7-8, Table 5, the labelling is unclear, especially ‘No of journals’ – please revise. Same with Table 7.

- p.7 lines 234-36, this highlights one of the weaknesses of this article, which is that it cites very little relevant research literature. The argument here would be strengthened by doing so.

- p.14 line 401 and line 415, replace ‘%2F’ with ‘/’ to aid with link resolving

- p.14 line 416, spelling error: ‘practice’

- p.14 line 429, the semi-colon should probably be a full stop

Author Response

(The authors gave the same response as above.)

Reviewer 4 Report

The actuality of the topic is very high, which is a positive value and it has been elaborated with a very short time period. On the other hand, I think that you should add relevant references in the theoretical framework. I'm following Plan S reports, articles, so I add my list below. Just consider some of them.

Plan S 

10 principles https://www.coalition-s.org/10-principles/  

Guidance on the Implementation of Plan S https://www.coalition-s.org/feedback/  

Funding bodies 

18.11.05 Wellcome  

Wellcome is updating its open access policy https://wellcome.ac.uk/news/wellcome-updating-its-open-access-policy  

Statements 

18.07.xx European Mathematical Society 

EMS on Open Access - Update July 2018 http://euro-math-soc.eu/news/18/07/10/ems-open-access-update-july-2018  

18.09.04 LIBER 

LIBER Supports New Plan to Make Open Access A Reality By 2020 https://libereurope.eu/blog/2018/09/04/liber-supports-new-plan-to-make-open-access-a-reality-by-2020/  

18.09.05 SPARC E:  

New coalition of European funders join together to place unprecedented mandate on researchers to publish OA https://sparceurope.org/new-coalition-of-european-funders-join-together-to-place-unprecedented-mandate-on-researchers-to-publish-oa/  

18.11.15. Young Academy of Sweden:  

Sweden Open letter on Plan S to the European Commission, Science Europe, governments and research funders within the EU https://www.sverigesungaakademi.se/openletterplan_s_yas  

18.11.00 Plan S Open Letter:  

Reaction of Researchers to Plan S: Too Far, Too Risky https://sites.google.com/view/plansopenletter/home    

18.11.11. JISC 

Working together to implement Plan S https://scholarlycommunications.jiscinvolve.org/wp/2018/11/27/working-together-to-implement-plan-s/ 

18.12.06. LIBER Open Access Working Group:  

Statement on Plan S Guidelines https://libereurope.eu/blog/2018/12/06/liber-statement-plan-s-guidelines/  

18.12.06. German Chemical Society on Plan S 

Statement of the German Chemical Society on Plan S https://www.gdch.de/fileadmin/downloads/Service_und_Informationen/Presse_OEffentlichkeitsarbeit/

18.12.13 COAR 

Building a Sustainable Knowledge Commons: COAR’s response to Plan S https://www.coar-repositories.org/activities/advocacy-leadership/open-science-and-sustainable-development/ 

19.01.08 (?) DARIAH EU 

Towards a Plan(HS)S: DARIAH’s position on PlanS https://www.dariah.eu/wp-content/uploads/2018/10/Towards-a-PlanHSS-excerpt.pdf  

Comments 

18.09.04 Nature 

Radical open-access plan could spell end to journal subscriptions: Eleven research funders in Europe announce ‘Plan S’ to make all scientific works free to read as soon as they are published / Holly Elsehttps://www.nature.com/articles/d41586-018-06178-7  

18.09.15 The economist 

European countries demand that publicly funded research be free: The S-Plan diet https://www.economist.com/science-and-technology/2018/09/15/european-countries-demand-that-publicly-funded-research-be-free?fsrc=scn/tw/te/bl/ed/europeancountriesdemandthatpubliclyfundedresearchbefreescientificpublishing 

18.10.02 Blog LSE  

The expansion of open access is being driven by commercialisation, where private benefit is adopting the mantle of public value http://blogs.lse.ac.uk/impactofsocialsciences/2018/10/02/the-expansion-of-open-access-is-being-driven-by-commercialisation-where-private-benefit-is-adopting-the-mantle-of-public-value/ 

18.11.26 The Scholarly Kitchen:  

Do You Have Concerns about Plan S? Then You Must be an Irresponsible, Privileged, Conspiratorial Hypocritehttps://scholarlykitchen.sspnet.org/2018/11/26/do-you-have-concerns-about-plan-s-then-you-must-be-an-irresponsible-privileged-conspiratorial-hypocrite/?informz=1 

18.11.11 For better science 

 Did Frontiers help Robert-Jan Smits design Plan S? / Leonid Schneider 

18.11.30 INNOVATIONS IN SCHOLARLY COMMUNICATION 

Nine routes towards Plan S compliance / by Jeroen Bosman & Bianca Kramer https://101innovations.wordpress.com/2018/11/30/nine-routes-towards-plan-s-compliance/  

18.12.05 INNOVATIONS IN SCHOLARLY COMMUNICATION  

Towards a Plan S gap analysis?  

(1) Open access potential across disciplines  

(2) Gold open access journals in WoS and DOAJ 

18.12.05 Scholarly Kitchen.  

Plan S: Impact on Society Publishers / By MICHAEL CLARKEDEC https://scholarlykitchen.sspnet.org/2018/12/05/plan-s-impact-on-society-publishers/?informz=1  

18.12.07 Scholarly Kitchen  

Plan S: A Mandate for Gold OA with Lots of Strings Attached / ANGELA COCHRANDEC https://scholarlykitchen.sspnet.org/2018/12/07/plan-s-a-mandate-for-gold-oa-with-lots-of-strings-attached/ 

18.12.11 APS News, December 2018 (Volume 27, Number 11) 

Plan S Tries to Flip the Open Access Switch, By Leah Poffenberger https://www.aps.org/publications/apsnews/201812/plan-s.cfm    

18.12.11 Open and shut? 

The OA Interviews: Peter Mandler  https://poynder.blogspot.com/2018/12/the-oa-interviews-peter-mandler.html?spref=tw  

18.12.12 Research Fortnight, 12 December 2018 

Plan S offers little for the people it will affect most / david Nicholas https://www.researchgate.net/publication/329584761_Plan_S_offers_little_for_the_people_it_will_affect_most  

19.01.03 Science 

Will the world embrace Plan S, the radical proposal to mandate open access to science papers? Science. By Tania Rabesandratana 

https://www.sciencemag.org/news/2019/01/will-world-embrace-plan-s-radical-proposal-mandate-open-access-science-papers  

19.01.04 Science Vol. 363, Issue 6422, pp. 11-12  

The world debates open-access mandates / Tania Rabesandratana DOI: 10.1126/science.363.6422.11    

19.01.04. David Wojick writings and stuff 

Plan S does not exist / By David Wojick, https://www.google.com/search?q=David+Wojick%27s+writings+and+stuff&oq=David+Wojick%27s+writings+and+stuff&aqs
=chrome..69i57j69i60j69i61&sourceid=chrome&ie=UTF-8  

Synopsis: Absent the APC cap, Plan S is decisively incomplete. 

19.01.06 Le blog de Bernard Rentier  

Mais quel est donc le statut du Plan S ? = What is the status of Plan S after all ?https://bernardrentier.wordpress.com/2019/01/06/mais-quel-est-donc-le-statut-du-plan-s/ 

By using DOAJ as the only data extraction source, having detected some of the weakness it has, I believe that the triangulation of results is necessary in order to be able to make firm conclusions. I recommend selecting a sample of journals and contrasting the DOAJ information on the magazine's own site, for instance.

Finally, making reference to the tables, you should look for an alternative way to present information that facilitates reading.

Author Response

(The authors gave the same response as above.)

Reviewer 5 Report

This article presents a novel result” the low-level of compliance of current Open Access journals with Plan S requirements. This result is both novel and of significant interest to readers and to the scholarly communication community in general. The methodology used has gaps, and would be made more robust with modest extensions, but is likely sufficient in its current state, to support the primary findings.

I recommend that the article be accepted for publication with three small but important revisions, noted below:

The discussion of the quality/accuracy of the DOAJ metadata set  in section 1 iis impressionistic and insufficiently precise. This could readily be remedied by direct validation of a small (25-50) random sample of journals, and reporting (in an appendix) the level of reliability found. This is a minor issue in this case, because given the patterns of compliance discovered, it is quite unlikely that the overall conclusions would be changed by even moderate levels of measurement error.  However, including a sample evaluation

As currently presented, there is a strong possibility that compliance is very substantially overestimated because four criteria are not evaluated (Criteria G, H, L and M  as documented in Table 2.)  While Criteria G and H are  unlikely to be a major source of noncompliance -- it is possible that only a small proportion of journals meet L and M -- which would substantially affect the conclusions of the article. Previous estimates of criterion L suggest that compliance is very low in OA journals  -- see
Evaluating and promoting open data practices in open access journals
E Castro, M Crosas, A Garnett, K Sheridan, M Altman - Journal of Scholarly Publishing, 2017. Previous baseline estimates should be recognized in section 3, and in the discussion in section 4.

Further, using a complementary small random sample analysis (as described above) would also provide estimates of overall (population) levels of compliance for these criteria. (I am not suggesting that a large sample be used to conduct the subgroup analysis.)

The presentation of the main results in Section 3 tables 5 and 6 conveys a level of certainty that is unwarranted. In fact each of these point percentages resulted are strictly false -- because of measurement error (the metadata does not exactly represent the current state of the journal; and the coding of the metadata is uncertain for some ambiguous cases); and ambiguity in interpretation of the Plan S requirements themselves.

The authors take a conservative approach (resolving ambiguity in the direction of non-compliance) in many cases (e.g. the measure of Criterion C; J);  but a non-conservative approach (resolving ambiguity in the direction of compliance) in others (e.g. Criterion F). As a result, it is impossible to interpret the sum of criteria percentages as either minimum or maximum -- nor to estimate the level of uncertainty associated with these numbers.

This is a serious methodological defect, but could readily be addressed by direct measurement of a small sample of journals (as discussed in 1, above) which could then be applied to estimating the uncertainies in all tables derived from these measures.

Author Response

Please see attached file where we have both general comments and replies to individual comments by the reviewers.

Round  2

Reviewer 1 Report

In brief: much improved but still needs work, particularly what appears to be an incorrect understanding of DOAJ's inclusion criteria which impacts your data (but not your results as the incorrect data are not the points where journals need help). Please see the attached for details. 

Reviewer 4 Report

The improvements made I think give more details to readers and  quality to the article.

Author Response

See attached file.

Round  3

Reviewer 1 Report

One major substantive point remains; see my review. I don't see this as a barrier to publication as it appears that PlanS itself is confusing on this point, but the article would be improved with a bit of work on this point. 
